# Microstructural and Mechanical Characterization of the Aging Response of Wrought 6156 (Al-Mg-Si) Aluminum Alloy

Nikolaos D. Alexopoulos [1,*], Joseph D. Robson [2], George Stefanou [1], Vasileios Stergiou [3], Alexandra Karanika [3] and Stavros K. Kourkoulis [4]

1 Research Unit of Advanced Materials, Department of Financial Engineering, School of Engineering, University of the Aegean, 41 Kountouriotou Str., 82132 Chios, Greece
2 School of Materials, The University of Manchester, Sackville Street Building, Manchester M1 3BB, UK
3 Hellenic Aerospace Industry S.A., 32009 Schimatari, Greece
4 Laboratory of Testing and Materials, Department of Mechanics, National Technical University of Athens, 9 Heroes Polytechnion Str., 15780 Athens, Greece
* Correspondence: nalexop@aegean.gr; Tel.: +30-2271035464

**Abstract:** The impact of the artificial aging response on the microstructure and tensile mechanical properties of aluminum alloy 6156 was investigated. Specimens were artificially aged at three different artificial aging temperatures and for various holding times to investigate all possible aging conditions, including the under-aged (UA), peak-aged (PA) and over-aged (OA) tempers. Microstructural investigation as well as tensile tests were performed immediately after the isothermal artificial aging heat treatment. An almost 50% increase in yield stress (around 340 MPa) was noticed in the PA temper and this was attributed to the precipitation of $\beta'$ and $Q'$ phases, consistent with the modelling predictions. This high yield stress value is accompanied by high values of elongation at fracture (>10%) that is essential for damage tolerance applications. The lack of large or interconnected grain boundary precipitates contributes to this high elongation. Slanted fracture was noticed for both UA and PA tempers, exhibiting a typical ductile and shear fracture mechanism. At the OA temper, coarsening of the precipitates along with broadening of the precipitate free zones resulted in a reduction in the strengthening effectiveness of the precipitates, and a small increase in the tensile ductility of approximately 12% was noticed.

**Keywords:** tension; artificial aging; microstructure; precipitates; fracture

## 1. Introduction

Aluminum alloys are widely used as the primary material for structural aircraft components due to their high stiffness-to-weight and strength-to-weight ratios; their high damage tolerance capabilities and resistance to corrosion offer additional advantages, when compared to other competitive materials. Nevertheless, the development of polymer matrix composites has driven aluminum alloy producers to develop new generation aluminum alloys that should be even lighter, with higher damage tolerance capabilities, and offering other functionalities, e.g., the possibility to be welded.

The production of welded components is gaining ground nowadays, where the manufacturing of integral components using the "traditional" manufacturing methods, e.g., milling, casting, etc., is not feasible or economically viable for the manufacturer. To this end, the ability of being efficiently welded, might prove to be an important factor for the wide-spread application of the structural material. Hence, aluminum alloy manufacturers focus their research on the weldability aspects of the under-developed aluminum alloys, since engineering applications utilizing welded aluminum structures are being exploited and are gaining ground against the competition.

Aluminum alloys (AA) from the 6xxx series (Al-Mg-Si), e.g., AA6056 were already selected in aircraft structures e.g., at the lower shell fuselage sections of the aircraft, where

damage-tolerance properties are required. Additionally, they exhibit an excellent compromise on corrosion resistance and high strength [1], thus making it particularly suitable for fuselage panels. The effect of thickness, geometry, and strain hardening behavior on the fracture mode for AA6056 was studied by Asserin-Lebert et al. [2] in relation to the precipitation microstructure. A quantitative TEM study has been performed by Delmas et al. in [3], showing that in the T6 condition, both needle-like and lath-shaped precipitates were formed. Shearing of the precipitates by dislocations was found to be the prevailing mechanism controlling the plastic deformation. In-situ TEM during straining tests performed by Delmas et al. in [4] revealed that the dislocation by-passing of the particles, assisted by cross-slip, is an additional mechanism that affects the work hardening ability of the alloy, mainly with the temperature increase. The precipitation kinetics of the hardening precipitates was experimentally investigated by Gaber et al. in [5], by means of differential scanning calorimetry, where the co-existence of $\beta'$ (associated with the ternary Al-Mg-Si system and stoichiometric analogy of $Mg_2Si$) and $Q'$ (associated with the quaternary Al-Cu-Mg-Si system and stoichiometric analogy $Al_5Cu_2Mg_8Si_6$) precipitates was reported. Schwerdt et al. in [6] performed fatigue studies on AA6056-T6, where the fatigue area fracture showed trans- and inter-granular cleavage-like planes. The inter-granular part was due to the precipitated free zones at the grain boundaries, while the trans-granular cleavage-like planes seem to be oriented at {110} crystallographic planes with the precipitates occupied.

The improved version of AA6056 is AA6156, which not only retains the excellent mechanical properties but also increases the damage tolerance capabilities. The increased damage tolerance is attractive for applications of 6156 sheet in aircraft fuselage, e.g., as reported in Lequeu et al. in [7]. Understanding of the structure-property relationships in AA6156 remains extremely limited; the effect of aging treatments on microstructure and hardness are reported in the literature, e.g., in [8–10], as well as the effects of Ag addition on the long-term thermal stability of the alloy, as reported by Zhang et al. in [11]. The high cycle fatigue performance was investigated by She et al. in [12], showing that the coarse particles on the surface or the interface between the second phase particles and the matrix play a key role during fatigue crack initiation. Likewise, the quench sensitivity of AA6156 toughness was also investigated by Morgeneyer et al. in [13].

The aging sequence in copper containing 6xxx alloys such as AA6156 is complex, but is generally considered to follow the findings for the Al-Mg-Si (containing Cu) alloys in the 1960s by Thomas in [14], such as SSSS (supersaturated solid solution) $\rightarrow$ GP zones $\rightarrow \beta''/\beta' + Q' \rightarrow \beta + Q$, where the $\beta$ phase and its metastable precursors are associated with the ternary Al-Mg-Si system (leading to $\beta$—$Mg_2Si$) and the equilibrium $Q$ phase is the quaternary $Al_5Cu_2Mg_8Si_6$. Whilst the exact sequence and phase designations remain the subject of debate, this combination of the two precipitate families is key to the good strengthening response of the alloy. Nevertheless, previous work by Tanaka and Warner in [15] on the similar, by chemical composition, AA6056 indicates that the major hardening phase in the T78 condition is the quaternary $Q$ phase. As demonstrated by Miao and Laughlin in [16], as well as by Esmaeili et al. in [17], the $\beta''$ phase might be considered as a precursor of both $\beta'$ and $Q$ phases in Al-Mg-Si alloys containing copper. Edwards et al. in [18] found that the $\beta''$ precipitates were found to be needle-shaped along <100>Al with a monoclinic structure, while in [19,20] the $\beta'$ precipitates were rod-shaped along <100>Al with a hexagonal crystal structure. Finally, $Q$ precipitates have a hexagonal crystal structure and form lath-like precipitates with the long direction lying along the <100>Al directions, e.g., [16] and [21].

The size and volume fraction of the formed precipitates have a profound effect on the mechanical properties of the material. As this aluminum alloy is widely used in the aircraft industry, it is of imperative importance to study the effect of artificial aging on the tensile mechanical properties of the alloy. Special care should be given to the mechanical properties that are considered for the design phase of the aircraft structures, such as yield stress and tensile ductility (elongation at fracture) that is often correlated with fracture toughness, e.g., Alexopoulos and Tiryakioğlu in [22]. The objective of the article is to investigate

the tensile mechanical properties of this aluminum alloy under various artificial aging conditions, and, to this end, an extensive database will be constructed. The investigation will include different aging temperatures and several different aging holding times. The tensile mechanical behavior of the new alloy AA6156 will be supported by microstructural investigations and such investigations, to the best of our knowledge, are still not reported in the literature for the specific alloy.

## 2. Materials and Methods

The material used in the present investigation was a commercially available 6156 wrought aluminum alloy in T4 condition provided by Constellium, that was received in sheet form with nominal thicknesses of 3.6 mm and without any surface corrosion protection (Alclad) layer. The chemical composition of the alloy (by weight percentage) was Si 0.7 to 1.3%, Mg 0.6 to 1.2%, Cu 0.7 to 1.1%, Mn 0.4 to 0.7%, Fe < 0.2%, Cr < 0.25%, Zn 0.1 to 0.7% and Al remainder. Tensile specimens were machined and prepared from the longitudinal (L) sheet rolling direction according to ASTM E8 specification with 12.5 mm × 3.6 mm being the reduced cross-section and 50 mm being the gauge length of the specimens.

Specimens for microstructural characterization and tensile testing were machined from the sheets. The non-heat treated (T4 condition refers according to the aluminum alloy heat treatment temper designations as solution heat treated, and naturally aged to a substantially stable condition) and heat-treated samples for metallographic analysis were prepared by standard grinding and polishing methods, with a final polish with oxide polishing slurry (O.P.S.) 1 μm. The samples were then etched with Keller's reagent before examination. Microstructural analysis was performed using a FEI Quanta 650 field emission gun scanning electron microscope (FEG-SEM) operated between 5 and 20 kV using backscattered electron imaging at a working distance of 4 mm. Phase fraction prediction and estimated precipitation times were calculated using JMatPro® version 9 (Sente Software) with the ALDATA thermodynamic database.

The tensile specimens were surface cleaned with alcohol and then artificially aged (heat treated) in an electric oven with air circulation Elvem (2600 W) with ±0.1 °C temperature control. Three different temperatures, namely 150, 170 and 190 °C, were exploited for artificial aging, while different aging holding times were chosen to be in the range between 1 and 166 h to investigate all aging conditions, including under-aged (UA), peak-aged (PA) and over-aged (OA) tempers, Table 1. The aging temperatures were selected to fill the whole temperature range of the artificial aging heat treatment of the commercial sheets of 6xxx aluminum series and closely to its upper bound, e.g., [23]. This high aging temperature, e.g., 190 °C, brings the alloy microstructure into the OA condition in a relatively short period of time.

**Table 1.** Artificial aging conditions of AA6156 and respective tensile specimens investigated.

| Temperature | 0 h (T4) | 1 h | 2 h | 4 h | 7 h | 9 h | 15 h | 24 h | 32 h | 48 h | 63 h | 98 h | 120 h | 166 h | No of sp. |
|---|---|---|---|---|---|---|---|---|---|---|---|---|---|---|---|
| 150 °C | | | | | | 3 | | 3 | | 3 | 3 | 3 | 3 | 3 | 21 |
| 170 °C | 5 | | 3 | 3 | 3 | | 3 | 3 | 3 | 3 | 3 | 3 | | | 27 |
| 190 °C | | 3 | 3 | 3 | 3 | | 3 | 3 | 3 | 3 | | | | | 24 |
| | | | | | | | | | | | | | | Total: | 77 |

All specimens were tensile tested immediately (no delay) after the artificial aging heat treatment to assess the effect of artificial aging on the respective tensile mechanical properties. Tensile tests were carried out in a servo-hydraulic Instron 100 kN testing machine according to ASTM E8 specification, with a grips constant deformation rate of $3.3 \times 10^{-4}$ s$^{-1}$. An attached extensometer was used the measure the displacement of the 50 mm gauge length at the reduced cross-section of the specimens during mechanical testing. A high elongation extensometer that could record high elongations up to 50% was used, as the alloy at the T4 temper exhibited high elongation exceeding 26% and the typical extensometers usually cover the range up to 10 or 20% elongation. Round-robin tests were

performed between the involved partners, namely the University of the Aegean, Hellenic Aerospace Industry, and the National Technical University of Athens. More than three specimens were tested in each different batch to obtain reliable average data, Table 1. A data logger was used during all tensile tests and the values of load, displacement and axial strain from the extensometer were recorded and stored in a computer. A total of more than 75 tensile stress–strain curves were analyzed for the evaluation of the tensile mechanical properties of the investigated AA6156 specimens.

## 3. Results

For the convenience of the reader, the microstructural and mechanical test results at the initial temper (T4 condition) will be reported first and then the effect of artificial aging follows.

### 3.1. Characterization of the Alloy at T4 Temper

#### 3.1.1. Microstructure

Microstructural analysis revealed that there are three classes of second phase particles present in the alloy. At the largest scale are the constituent particles, formed during casting and modified during homogenization, rolling, and solution treatment. Examples of constituents can be seen in the FEG-SEM micrograph—in backscattered mode (BSE)—in Figure 1a. These particles are typically up to around 25 μm in diameter and consist mainly of iron- and manganese-rich intermetallic compound (predicted to be mainly $Al_6(Fe, Mn)$). Also visible within the grains, but at a much finer scale, are the dispersoid particles that are precipitated during homogenization and used to control the grain structure. These are predicted to be predominantly of the $\alpha$-Al(FeMnCr)Si type. A higher magnification image showing several of these dispersoids in more detail is given in Figure 1b. Similar results were also found by Morgeneyer et al. in [13] regarding the dispersoids, which usually have dark color and equiaxed dimensions. It is also clear that dispersoids serve as nucleation points for large, elongated precipitates. Not visible in the micrograph is the third class of particles, the strengthening precipitates, which in the T4 state are expected to consist only of the nanoscale GP zones. JMatPro® estimates the volume fraction of GP zones formed at room temperature in this alloy to be around 1.1%.

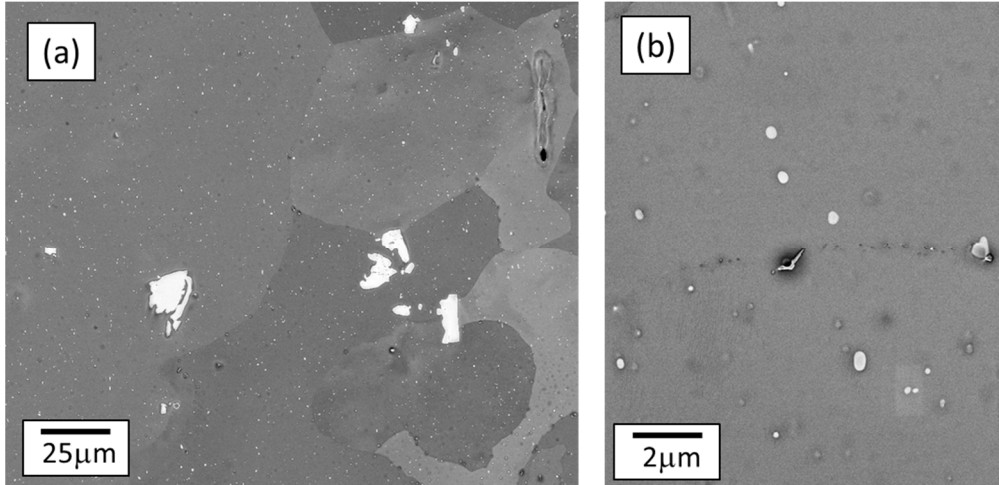

**Figure 1.** BSE imaging mode microstructure of AA6156 in T4 condition: (**a**) low magnification, showing large constituent particles with fine dispersoids just visible within the grains and (**b**) higher magnification image showing details of the dispersoids.

#### 3.1.2. Tensile Mechanical Properties

Figure 2 shows typical experimental tensile curves of reference specimens; AA6156 in the T4 temper can be classified as a low-strength and high-tensile ductility aluminum alloy

(Table 2). The evaluated test results showed that the reference specimens at T4 condition exhibited high elongation at fracture $A_f$ values, and even higher than 25%. Morgeneyer et al. in [13] performed synchrotron radiation computed tomography for AA6156 and the results showed failure ahead of the crack tip of the grain boundaries at 45° to the loading axis. Fracture was predominately intergranular, while both voiding and shear decohesion was noticed. Yield stress $R_p$ values of AA6156 were found to be at low levels (approximately $R_p$ = 230 MPa) when compared to other wrought aluminum alloys used in aeronautics, e.g., from 2024-T3 (yield stress $R_p$ = 385 MPa) from Alexopoulos et al. in [24] and from 2198-T851 ($R_p$ = 432 MPa) as reported in Alexopoulos et al. in [25]. Nevertheless, similar $R_p$ values for the T4 temper were noticed by Liu et al. in [26]. Ultimate tensile strength $R_m$ was found to be essentially higher than yield stress ($R_m$ = 338 MPa), while the results from Liu et al. in [26] showed approximately similar tensile mechanical properties.

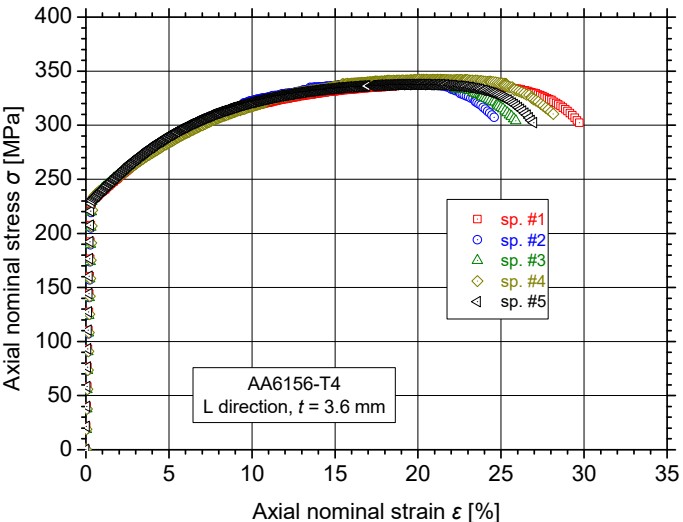

**Figure 2.** Typical five experimental tensile curves of aluminum alloy 6156 specimens at T4 condition.

Worth noting is also the high elongation at fracture $A_f$ values of this alloy at the T4 condition (exceeding 26%) was unexpectedly high. The test results on the base material (T4 condition) were triple-checked by internal round-robin tests to confirm the test results. It must be pointed out that Ren et al. [27] reported even higher elongation at fracture $A_f$ value for the T4, slightly higher than 30%. The data for the elongation at fracture of this alloy at the T4 temper seems to be extremely limited and, to this end, the relevant material properties data at different temper (e.g., T6) was also included in Table 2 for comparison purposes. The value of elongation at fracture at T6 temper ranges from 12.5%, according to Zhang et al. in [11], up to 19.5%, according to Lin et al. in [8]. Li et al. [28] refer to artificial aging for 8 h at 180 °C to obtain T6 condition from the starting T4 temper with hardness values of approximately 128 HV. Viscousi et al. [29] reported that the T6 condition is being resolved by artificially aging for 16 h at 175 °C from the starting condition (T4 temper). In a relevant article by the authors [24], the diffusion rate of different elements (e.g., Mg and Cu) on the aluminum matrix was investigated. The maximum and minimum diffusion rates of these elements on the aluminum matrix were calculated and it was shown that the empirical practice, when the aging temperature increases by 10 °C the diffusion rate is multiplied with a factor of two, is actually in between the maximum diffusion rate for Cu and the minimum diffusion for Mg for Al-Cu-Mg-Si alloys. Hence, by taking this semi-empirical guideline, it is estimated that that the equivalent aging time of 16 h at 175 °C is approximately 24 h at 170 °C. Linear regression between the experimental data (15 h and 28 h) was used to extract the mechanical properties of the specific aging temperature and the results are also shown in Table 2. The data from the literature are rather confusing regarding the heat treatment conditions required to achieve T6 condition, i.e., a shorter time than the one proposed by Viscousi et al. in [29] to achieve T6 condition (8 h at 175 °C)

was proposed by Lin Li et al. in [8] and Ren et al. in [27]. Additional data results from the literature, e.g., by Zhang et al. in [11], were gathered and provided in the same Table, where small discrepancies in tensile mechanical properties are noticed from the experimental test results.

**Table 2.** Average tensile mechanical properties of modulus of elasticity $E$, yield stress $R_p$, ultimate tensile strength $R_m$ and elongation at fracture $A_f$ of AA6156.

| Alloy/Temper | Source | $E$ [GPa] | $R_p$ [MPa] | $R_m$ [MPa] | $A_f$ [%] |
|---|---|---|---|---|---|
| 6156-T4 | Experimental results | 68.337 ± 0.867 | 230 ± 4 | 338 ± 3 | 26.5 ± 1.3 |
| 6156-T4 | Liu et al. [26] | - | 230 | 341 | - |
| 6156-T4 | Ren et al. [27] | | 179 | 301 | 30.3 |
| 6156-T6 | Viscousi et al. [29] | 69 | 341 | 378 | - |
| 6156-T6 | Lin et al. [8] | - | 341 | 426 | 19.5 |
| 6156-T6 | Zhang et al. [11] | - | 314 | 369 | 12.5 |
| 6156-T6 | Ren et al. [27] | | 211 | 341 | 15.4 |
| 6156-T6 (extrapolation of artificial aging of 20 h at 170 °C) | Experimental results | - | 332 ± 7 | 380 ± 4 | 13.1 ± 0.5 |

### 3.2. Effect of Artificial Aging

#### 3.2.1. Microstructure

The aging sequence in copper containing 6xxx aluminum alloys such as 6156 is complex but is generally considered to follow SSSS (supersaturated solid solution) → GP zones → $\beta''/\beta' + Q' \rightarrow \beta + Q$ [30], where the $\beta$ phase and its metastable precursors are associated with the ternary Al-Mg-Si system (leading to $\beta$—$Mg_2Si$ [31]) and the equilibrium $Q$ phase is the quaternary $Al_5Cu_2Mg_8Si_6$. Whilst the exact sequence and phase designations remain the subject of debate, this combination of two precipitate families is key to the good strengthening response of the alloy. Zhang et al. [11] found that at low Cu concentration (<0.25 wt.%) the $\beta$-type phase is being formed while for Cu > 2.5 wt.% the $Q$-type precipitates are formed. Hence, in the region of 1.0 wt.% Cu, a proportion of $\beta + Q$ precipitates is expected. Yin et al. [32] in a similar—by chemical composition—aluminum alloy (Al—0.42 Mg—0.5 Si—1.0 Cu) measured the average needle length of the precipitates to be approximately 12.3 ± 0.4 nm. Transmission electron microscopy (TEM) studies in [3] indicated that the $Q'$ phase forms as laths, whereas the $\beta''$ phase forms as rods, and this combination is expected to provide an effective barrier to dislocation motion, e.g., [33,34]. The FEG-SEM method in backscattered mode (BSE) used in the present work is not able to resolve the strengthening precipitates in the UA and PA condition, and so the expected phases were predicted using JMatPro®. Figure 3a shows the predicted phase fractions of the metastable phases that can form in 6156, where both $\beta'$ and $Q'$ phases are predicted, with the $\beta'$ dominating. The kinetics of the precipitate can also be predicted by JMatPro® and Figure 3b shows a time–temperature–transformation diagram for the onset of precipitation of the metastable phases in the present case, superimposed on which are the time ranges used in the present study at the three aging temperatures (150, 170, 190 °C). The whole time and temperature range, as well as both $\beta'$ and $Q'$, are predicted to be present. At the longest over-aging (OA) times, the transformation to equilibrium $\beta$ ($Mg_2Si$) phase is predicted to have started.

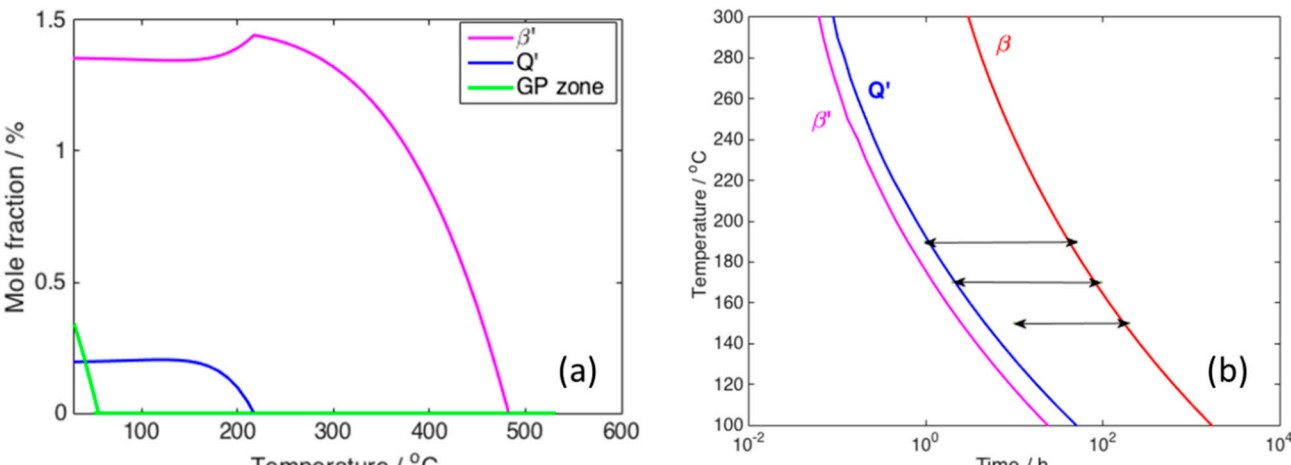

**Figure 3.** (**a**) Predicted metastable phase fractions at completion of precipitation as a function of aging temperature and (**b**) predicted onset of precipitation of metastable and equilibrium *β* phase. Also shown are the aging ranges used in the present work.

Figure 4 in BSE imaging mode shows a general view of the UA microstructure. The average grain size is below 100 μm and, as expected, the grain structure is unchanged from the T4 condition. The constituents and dispersoids present in the T4 condition are also visible and unchanged, as expected. The inset image shows a higher magnification view of a grain boundary. The grain boundary is free of large precipitates, but a low number of very small grain boundary precipitates are visible.

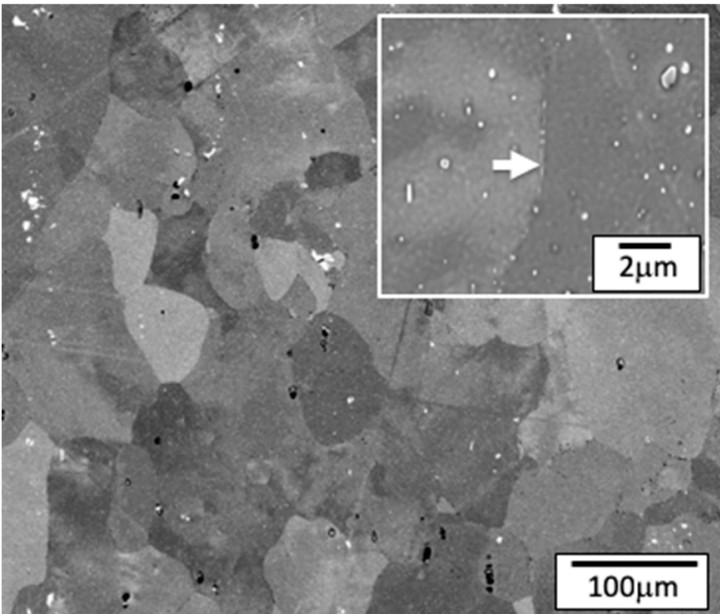

**Figure 4.** Typical images of the BSE imaging mode microstructure of AA6156 in the under-aged (UA) state (24 h at 150 °C). Low magnification showing grain structure and constituent particles (white regions). Insert: Higher magnification showing dispersoids within the grains and a grain boundary (arrow) with few grain boundary precipitates.

Figure 5 shows the respective microstructural images of the alloy at the PA condition. The low magnification image reveals no microstructural change at this scale, as expected. The higher magnification inset shows a grain boundary triple point, and the three grain boundaries radiating from this point are decorated in small grain boundary precipitates. The extent of this precipitation is different on each boundary (e.g., the boundary below

the arrow has more precipitation). However, even in this temper, the grain boundary precipitates remain very small (<20 nm in equivalent diameter), and most of the grain boundary area is not decorated. The relatively low fraction and size of the grain boundary precipitates is important to understanding the fracture behavior of this alloy and will be demonstrated later.

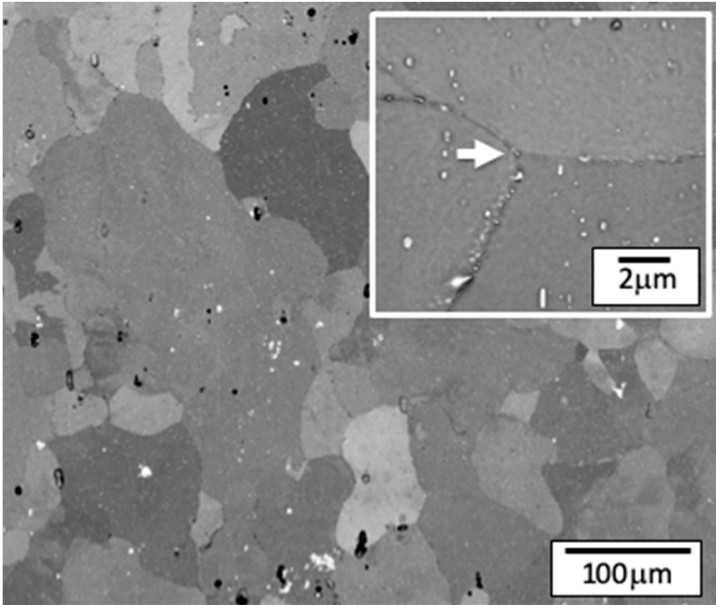

**Figure 5.** BSE imaging mode microstructure of AA6156 in the peak-aged (PA) state (28 h at 170 °C) showing grain structure and constituent particles. Insert: Higher magnification showing dispersoids within the grains and a grain triple junction; evidence of small precipitates on the grain boundaries.

The precipitates in the OA condition are sufficiently large to be imaged in the FEG-SEM, and examples of the OA microstructure at low and high magnification are shown in Figure 6. High aspect ratio precipitates in the grain interior, with lengths up to approximately 50 nm can be seen, with precipitate free zones formed adjacent to the grain boundaries, as also noticed in [35]. Similar results have been noticed in Yin et al. [32], where the needle lengths exceeded 50 nm and lengths up to 72.5 nm were also measured. An increase in the amount of grain boundary precipitation was also observed, and examples are shown in Figure 6c. These precipitates often do not lie along the grain boundary, but instead have well-defined orientations with respect to one of the grains. Li et al. [28] reported that in the over-aging condition the $Q'$ precipitates increase their volume fraction and coarsen. Additionally, grain boundary precipitates also coarsen and are sparsely distributed, while several PFZs appears. Extreme OA has an impact on the broadening of the PFZs. Similar results were noticed by Tanaka and Warner [15] regarding the quaternary precipitations in AA6056 when comparing them at the PA and OA conditions. As expected, the constituents and dispersoids are unaffected by the aging process.

### 3.2.2. Mechanical Properties

Typical engineering stress–strain tensile flow curves can be seen in Figure 7 for the three investigated isothermal artificial aging temperatures. The tensile flow curves are different for those specimens with higher artificial aging exposure (duration) for the same aging temperature, e.g., at 150 °C, as presented in Figure 7a. Ultimate tensile strength $R_m$ is continuously increasing with increasing aging time up to the latest investigated time of 166 h, while a simultaneous drop in ductility (elongation at fracture $A_f$) is noticed.

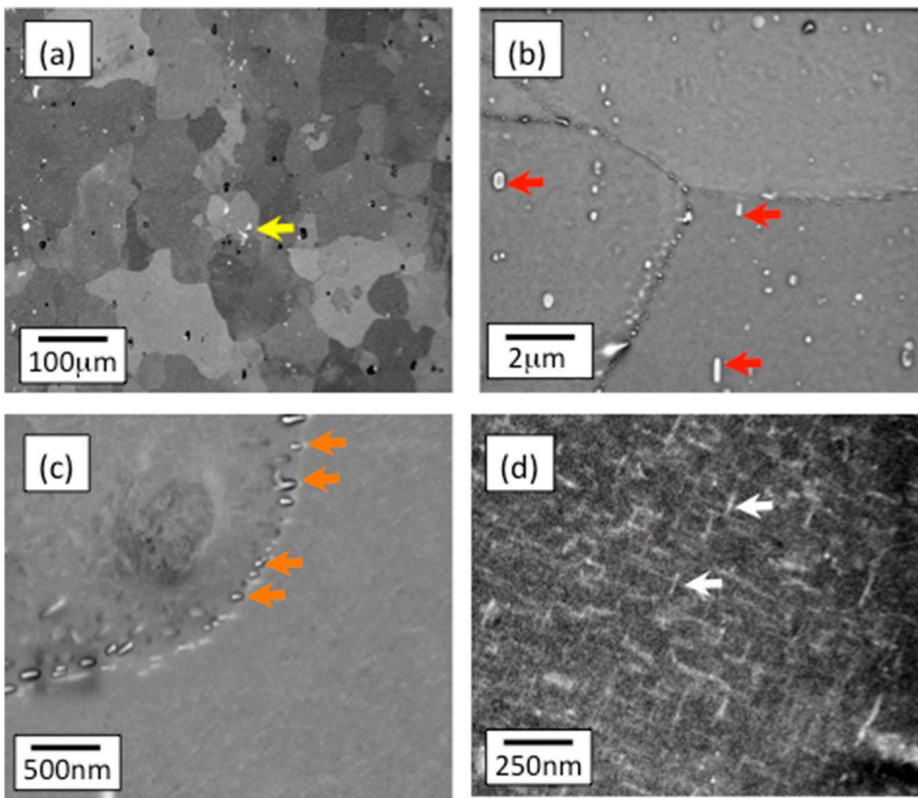

**Figure 6.** BSE imaging mode microstructure of AA6156 in the over-aged (OA) state (40 h at 190 °C). (**a**) Low magnification showing grain structure and constituent particles (arrowed) (**b**) intermediate magnification, showing dispersoids (arrowed), (**c**) high magnification showing a grain boundary decorated with precipitates (arrowed), (**d**) very high magnification showing the overaged strengthening precipitates within a grain (arrowed).

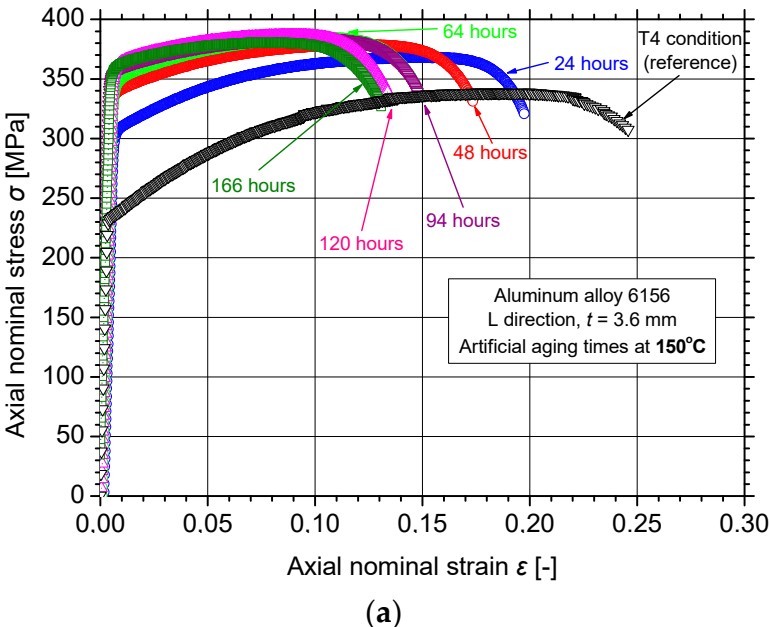

(**a**)

**Figure 7.** *Cont.*

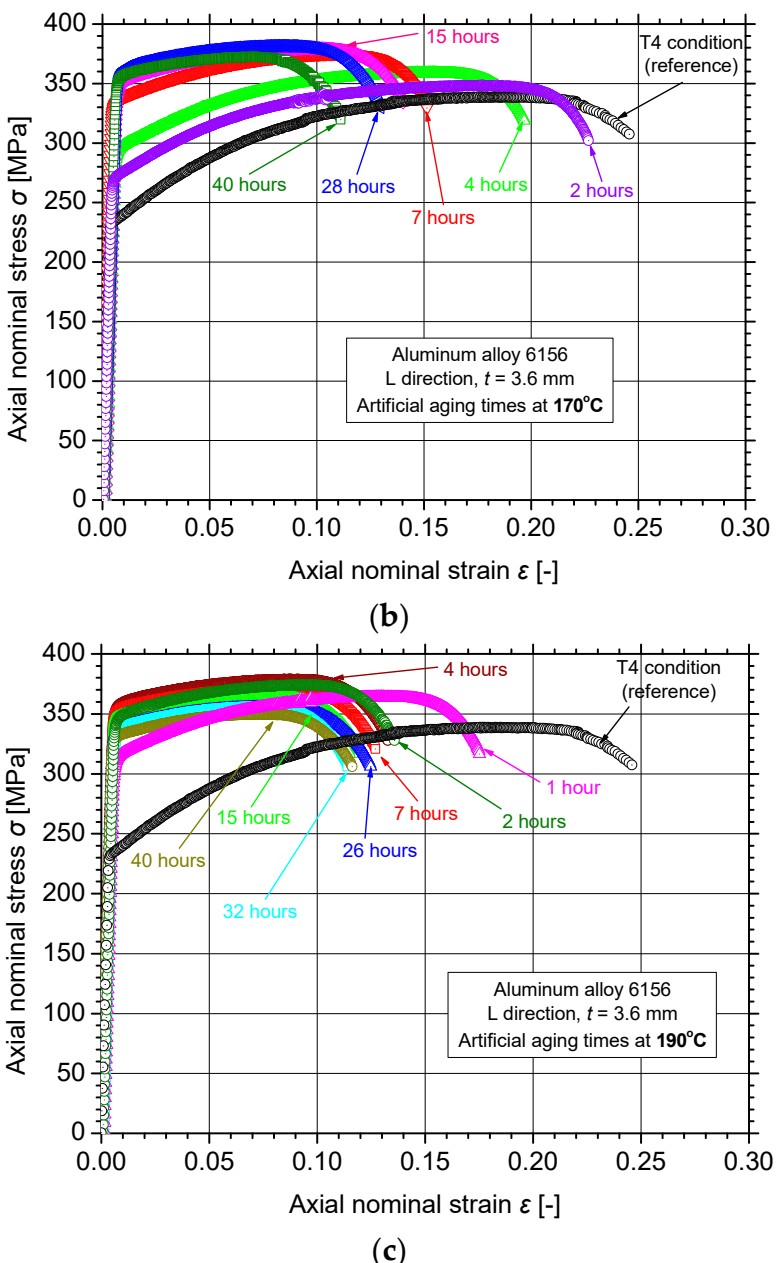

**Figure 7.** Typical experimental tensile flow curves of AA6156 specimens for different heat treatment temperatures (**a**) 150 °C, (**b**) 170 °C and (**c**) 190 °C.

Figure 7b shows the respective tensile flow curves for the investigated aging temperature of 170 °C. Ultimate tensile strength and yield stress is increasing with increasing aging time up to a maximum of 28 h; additionally, a simultaneous drop in elongation at fracture is well noticed. For longer aging periods (>28 h), a gradual decrease in the previously mentioned strength properties is recorded. A partial recovery of the tensile ductility was noticed for the specimens after the PA regime, e.g., for aging times higher than 48 h.

The same trend of the tensile flow curves is also noticed for the higher investigated aging temperature of 190 °C (Figure 7c). In the following, the evaluated mechanical properties of yield stress $R_p$, ultimate tensile strength $R_m$ and elongation at fracture $A_f$ will be individually discussed.

Figure 8a shows the variation of yield stress $R_p$ for different aging times and different aging temperatures as average values along with the respective standard deviation. The non-artificially aged specimens (T4 condition) exhibited yield stress $R_p$ = 230 MPa which

increased with increasing aging time up to a maximum value; this maximum value was found to be aging-temperature-dependent. For the lower investigated temperature, this peak was not achieved and for time durations higher than 166 h the alloy is expected to slightly increase its yield stress above 340 MPa. Hence, all investigated time durations of this aging temperature were related to UA and PA conditions, since no OA signs (drop in ductility) were noticed. The tensile test results regarding artificial aging of AA6156 are very limited in the literature. Zhang et al. [11] reported that a peak in conventional yield stress $R_{p0.2\%}$ was noticed of around 320 MPa after aging 100 h at 150 °C. With OA conditions (1000 h at 150 °C) the conventional yield stress decreases at the minimum plateau of values of approximately 310 MPa.

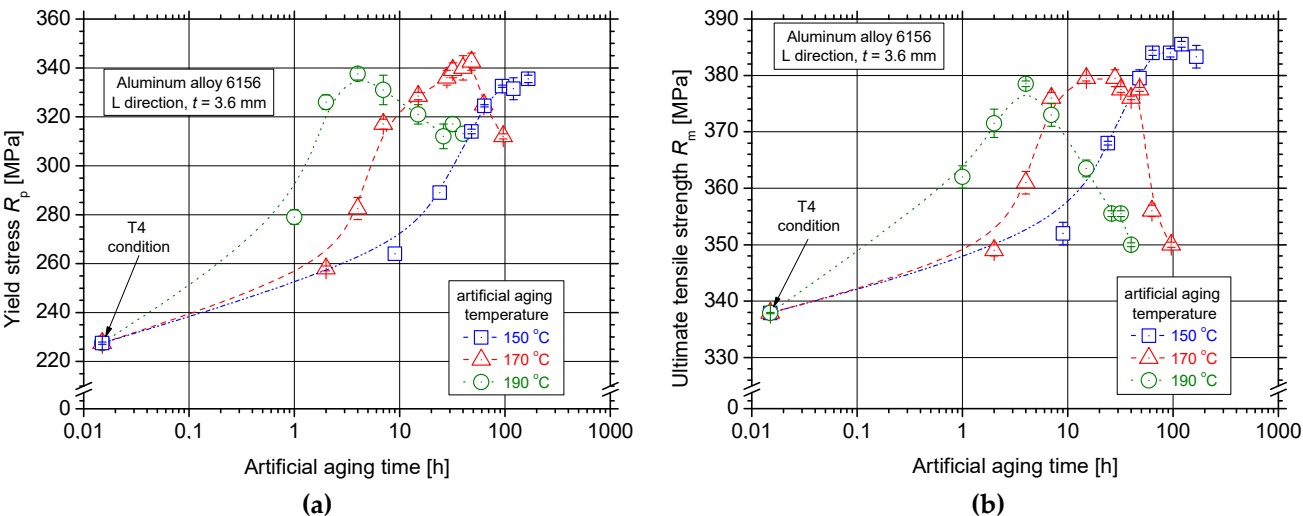

**Figure 8.** Evaluated tensile test results of (**a**) yield stress $R_p$ and (**b**) ultimate tensile strength $R_m$ for the different investigated artificial aging conditions of AA6156.

When comparing the results of the 170 °C and 190 °C aging temperatures, the peak yield stress was shifted to lower aging time (48 h at 170 °C and 4 h at 190 °C) and lower $R_p$ value (343 MPa at 170 °C and 337 MPa at 190 °C), when artificially aging the material at higher temperature. In [24], it was shown that for the case of AA2024-T3, peak yield stress was found to be dependent on the iso-thermal aging temperature and it was experimentally noticed that a 6 MPa yield stress drop exists for every 10 °C increase in aging temperature. It is well established, e.g., [14], that with the lower aging temperature–higher peak strength combination can be achieved due to the well-balanced formation of coherent and subsequent non-coherent $\beta$ and $Q$ type precipitates, respectively. According to [23], from PA and after, no more second phase is precipitated, while growth of the precipitates is observed, leading to a smaller number of larger precipitates. For the higher temperature of the artificial aging heat treatment, only $Q'$ precipitates are expected to be formed, e.g., in [30].

The same concept as above seems to be the case for the ultimate tensile strength $R_m$ results presented in Figure 8b. Slightly lower peak $R_m$ values were noticed for the different aging temperatures, proving that, unlike $R_p$, ultimate tensile strength is less aging temperature sensitive. Additionally, in [24] it was shown that for the diffusion of Cu and Mg atoms in the aluminum matrix to form the *S*-type precipitates with chemical composition $Al_2CuMg$, the empirical practice of «equivalent» aging time to be reduced twice when increasing the aging temperature by 10 °C was followed for AA2024. The diffusion of Mg and Si atoms in an aluminum matrix will be performed in a follow-up article to investigate whether the above empirical rule is applicable to the investigated alloy.

Figure 9 shows the variation of elongation at fracture $A_f$ for different aging times and temperatures. As expected, with increasing artificial aging time, ductility decreases continuously till a «plateau» is reached; this plateau lies within the range of 11 and 13%

elongation at fracture values. This plateau was also noticed by Zhang et al. [11], where $A_f$ remained almost constant and around 12% for the various aging times up to 1000 h at 150 °C. As expected, the decrease rate of elongation at fracture is higher for the higher aging temperatures investigated. An inverse behavior can be noticed for elongation at fracture against the respective yield stress values up till the PA condition (assuming to be at the peak yield stress point of Figure 8a). For the OA specimens at 170 °C aging temperature (higher aging times than 12 h), some limited recovery of the elongation at fracture is noticed at the expense of the yield stress decrease. This small recovery was also reported for the specimens at 190 °C aging temperature from 2 h up to 8 h aging; nevertheless, for the extreme OA condition the elongation at fracture seems to be consistent in the «plateau» of minimum values, ranging between 11% and 13%.

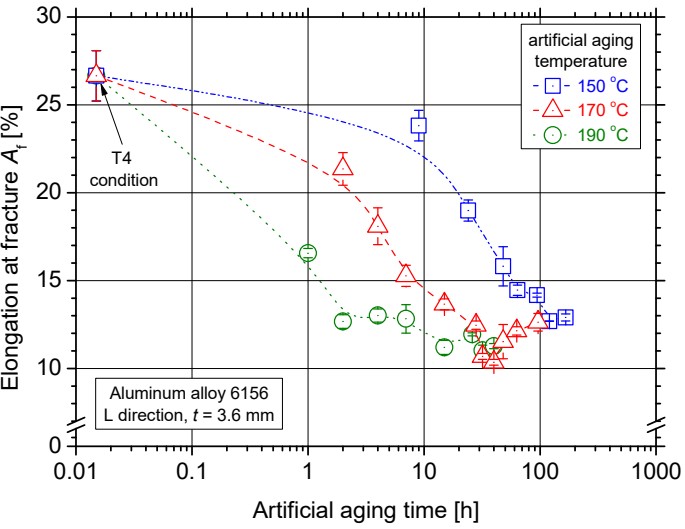

**Figure 9.** Evaluated elongation at fracture $A_f$ for the different investigated artificial aging conditions of AA6156.

## 4. Discussion

Figure 10 shows the cross-plotting of two tensile mechanical properties over different aging times and temperatures. Yield stress $R_p$ was selected to account for the strength properties and elongation at fracture $A_f$ for ductility and, overall, for the plastic deformation ability. The authors place emphasis on these two mechanical properties as yield stress is an essential mechanical property that is considered when designing lightweight structures, while the high elongation value is an essential property that in most cases is analogous to the damage tolerance capabilities of the alloy, e.g., fracture toughness and fatigue crack growth. It is evident from the figure that for the UA conditions, elongation at fracture decreases in an analogous (linear) behavior to the yield stress increase. The region at PA conditions seems to sacrifice tensile ductility (almost 2% elongation at fracture) on small-gained strength value, namely less than 10 MPa. In the case of OA condition there seems to be a linear trend for increasing ductility over decrease in yield stress.

The changes in mechanical properties are dictated largely by the changes in precipitates volume fraction and their distribution. In the T4 state, the microstructure consists of the expected constituent and dispersoid phases, with GP zones also predicted to be present. On aging, the precipitates evolve, following a complex sequence as previously discussed. It is predicted that across the range of aging time studies, the microstructure will contain two families of strengthening precipitates, $\beta'$ [34] and $Q'$ [36]. However, in the UA condition, these precipitates will be in their early stages of formation with a low volume fraction and will not provide the most effective barrier to dislocation motion. As the precipitates evolve, their strengthening effect increases and the dislocations to be moved either need to shear or to bypass the evolved precipitates. The effect of artificial aging on several precipitation-hardened aluminum alloys can be found in several articles focused on the description of

the precipitations and their volume and size changes with increased aging time, e.g., [37]. The effect of the microstructural characteristics (volume and size) of the precipitates on the strength properties, i.e., yield stress, are well reported in the literature [38]. Nevertheless, for the specific Al-Mg-Si alloy, it is predicted that the increase in yield stress is not associated with a change of phase, but rather the progression of the $\beta'$ and $Q'$ precipitation. At the longest aging times, as the precipitates coarsen, their strengthening effectiveness reduces (OA) and it is also predicted that transformation to the equilibrium $\beta$ phase has begun. At this stage, the precipitates are large enough to resolve using FEG-SEM. There is also evidence of extensive precipitation at grain boundaries in the OA state, which are likely to have a deleterious role on tensile ductility. These precipitates often form as aligned plates or rods that rather than lying along the grain boundary are oriented along a particular plane in the adjacent matrix.

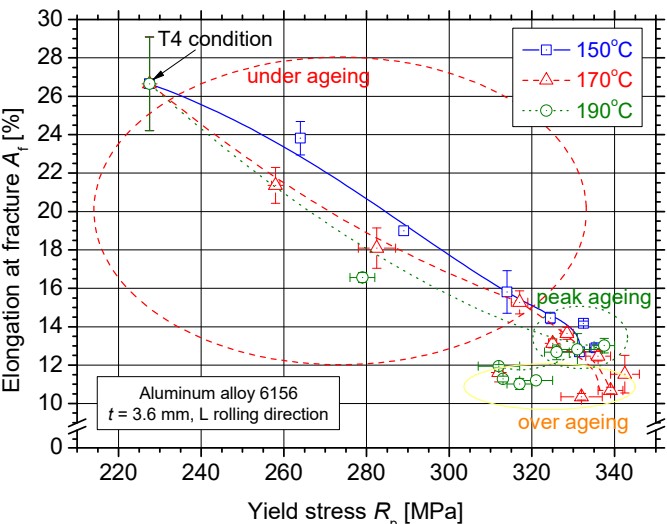

**Figure 10.** Plotting map of tensile elongation at fracture $A_f$ over yield stress $R_p$ for the different artificial aging conditions of AA6156.

## 5. Fractography

Side fracture morphologies of AA6156 tensile test samples for all the investigated aging conditions can be seen in Figure 11. As can be seen in the first three figures, the fracture mechanism is identical for the UA and PA conditions. This involves a purely ductile fracture mechanism and shear (slanted fracture) that is demonstrated by the 45° angle with the loading axis (Figure 11a–c). This kind of localization angle of 45° is characteristic of plane strain state, despite of the low thickness of 3.6 mm of the specimens. Necking occurs preferentially along the thickness direction and therefore the deformation along the transverse direction is inhibited. Therefore, plane strain conditions are applied at the center of the neck that leads to slant fracture (45° angle) with the normal to the fracture plane. Similar results were noticed by Asserin-Lebert et al. [2] for several thicknesses ranging from 1.4 to 6.0 mm as well as by Morgeneyer et al. [13] for 3.2 mm thickness.

SEM analysis of the fracture surfaces (not shown here) showed several dimples which were also evidence of the ductile fracture mechanism. Ren et al. [27] also noticed several dimples and of different sizes, thus implying that the size of the precipitates varied significantly. Asserin-Lebert et al. [2] noticed that in the slanted regions of the T4 condition, dimples of several sizes were formed that were categorized on (fractured) coarse precipitates, dispersoids and $Q'$ precipitates. The large dimples were very limited, thus proving that coarse precipitates do not control the fracture mechanism in the T4 condition. The heat-treated samples (over-aged) had a fracture surface that consisted of dimples from coarse precipitates and dispersoids and therefore a change in the fracture mechanism was noticed. Likewise, Morgeneyer et al. [13] reported voids of the order of 10–40 μm on

the fracture surfaces and they were associated with coarse second phase particles, with a relatively small amount of final coalescence via finer, secondary void formation.

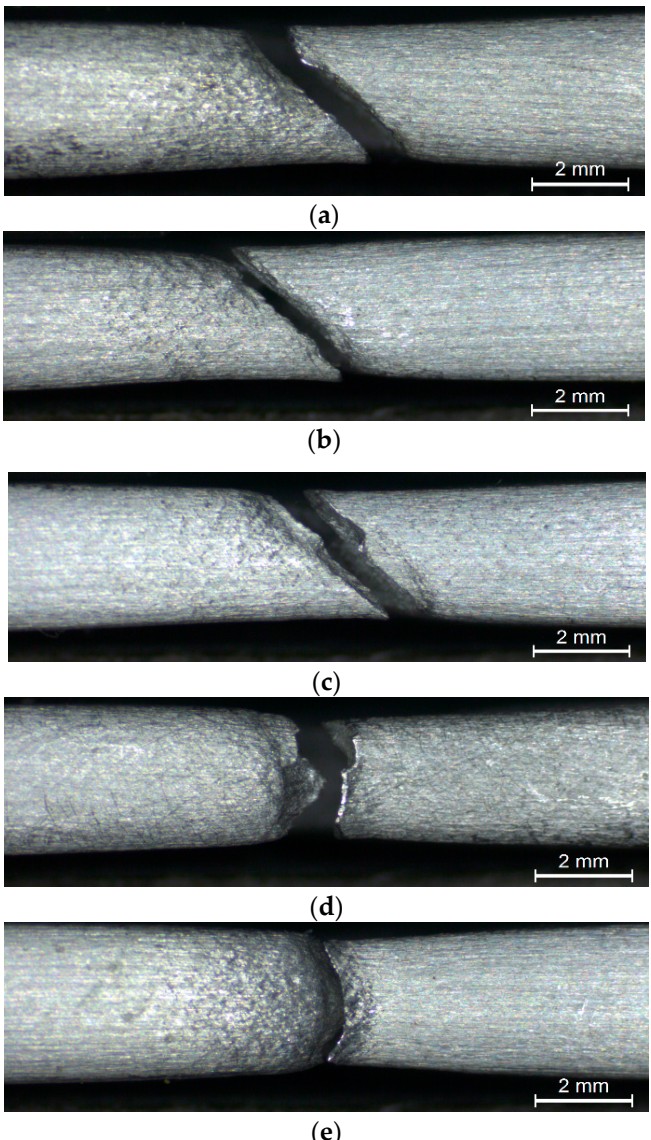

**Figure 11.** Side fracture surfaces of the tensile specimens with different artificial aging times (**a**) 7 h (under-aged—UA), (**b**) 28 h and (**c**) 48 h (peak-aged—PA) and finally (**d**) 63 h and (**e**) 96 h (over-aged—OA) at 170 °C.

For the case of OA condition (Figure 11d,e), the fracture mechanism continues to be ductile (also supported by high deformation—exceeding 12% in elongation at fracture), nevertheless this 45° angle with the loading axis is no longer existing. The fracture mechanism at the OA condition is macroscopically flat and reminds of a cup-and-cone geometry alike with rounded specimens. In [32], tearing ridges were also noticed in this aging condition and the SEM analysis revealed several broken particles in the center of the tough dimples. Hence, it was implied that the crack formation is related to the coarse phases that dominate fracture and does not allow for the shear mechanism to appear.

## 6. Conclusions

The following conclusions can be drawn from the present article:

(1) The microstructural investigation of AA6156 revealed that three types of particles can be observed in the T4 condition: coarse particles, dispersoids and strengthening precipitates that are assumed to be GP zones at the nanoscale. The volume fraction of GP zones was calculated to be around 1.1%. Elongation at fracture of AA6156 at the T4 condition exceeds 26%, which is essential for damage tolerance applications.

(2) Precipitates in the UA and PA condition in the grain interior are too small to image in the FEG-SEM but are calculated to consist of a mix $\beta''/\beta'$ phases and $Q'$ phases, with an approximately 6:1 volume fraction ratio. In the PA condition, precipitation leads to a strength increase of 110 MPa, resulting in a yield stress of 340 MPa.

(3) OA leads to precipitates, both within the grains and on the grain boundaries. The precipitates within the grains are predicted to be a mixture of $Q'$, $\beta'$ and $\beta$. Precipitates on the grain boundary remain very small (<20 nm in size even after OA).

(4) Isothermal artificial aging increases yield stress and decreases elongation at fracture up until PA condition. A linear correlation between the increase in yield stress and decrease in elongation at fracture was established for the different aging temperatures.

(5) For all investigated aging conditions, tensile elongation at fracture was not less than 10 to 13%, which is essential for high damage tolerance applications.

(6) The tensile fracture mechanism is identical from the initial temper (T4) till the PA condition, where slant fracture occurred; this corresponds to ductile fracture mechanism and shear. At the OA condition, only the typical void-coalescence fracture mechanism occurred (without 45° angle with the normal to the fracture plane) and it was attributed to the formation of cracks related to the coarse precipitates.

**Author Contributions:** Conceptualization, N.D.A.; methodology, G.S. and V.S.; validation, N.D.A., A.K. and S.K.K.; resources, N.D.A., A.K. and S.K.K.; writing—original draft preparation, N.D.A. and J.D.R.; writing—review and editing, N.D.A., J.D.R. and S.K.K.; All authors have read and agreed to the published version of the manuscript.

**Funding:** This research received no external funding.

**Data Availability Statement:** Available upon request.

**Conflicts of Interest:** The authors declare no conflict of interest.

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
