# Peer review of "Microstructural and Mechanical Characterization of the Aging Response of Wrought 6156 (Al-Mg-Si) Aluminum Alloy"

_alloys, doi:10.3390/alloys1020011_

Round 1

Reviewer 1 Report

1. Authors may consider merging Figure 4 and 5.

2. Figure 4, 5 and 6. Which mode is used SEM investigation? BSE or SE?

3. In Table 2 reference is missing.

4. Establish phase change through XRD for artificially aged samples.

5. Authors must round-off the values mentioned in the Table 2.

6. Fractograph of the failed samples must be shown.

Author Response

See attached electronic file with the step-by-step response to the reviewer

Reviewer 2 Report

Until recently, aluminum alloys were a very attractive material in the aerospace industry. Currently, the proportion of this material is decreasing in favor of composite materials. Nevertheless, aluminum alloys are still classified as modern and very attractive construction materials, this is due to the low specific weight and high specific strength. Hence, it is important to constantly develop this group of materials. The authors attempted to analyze the influence of aging parameters (temperature and time) on the mechanical properties of the 6xxx series alloy. The work is an interesting contribution to the development and understanding of the mechanisms responsible for shaping the properties of precipitation hardening aluminum alloys. The work is interesting, but it does contain some inaccuracies:

1. Lack of information whether the investigated alloy is commercial or laboratory (it may suggest the presentation of the ranges of element concentrations), however, it is worthwhile to carefully analyze the echo composition of the alloy using the spectroscopic method.

2. Line 122 - what does OPS mean?

3. It is worth giving the conditions of T4 treatment

4. Table 2 - bibliography error has occurred

5. Line 186 - How was the yield strength - 0.2% determined?

6. Was the PLC (Portevin-Le Chatelier effect) phenomenon observed during the static tensile test in the T4 state?

7. In line 403 the authors suggest the presence of shear mechanisms and the formation of dislocation loops. I understand that the authors mean the Motto-Nabarro mechanism and the Orowan-Ashby mechanism. The former occurs with small and coherent secretions, the latter with larger secretions. Do the authors suggest the presence of both mechanisms simultaneously?

8. In the discussion, a lot of space was devoted to the analysis of particle separations. However, it barriers to the work of own research with the use of TEM. They would make it possible to unequivocally define the type of partitions, their size and dispersion. Moreover, the authors suggest the presence of GP zones. These structures are very difficult to observe even with TEM. It is difficult to agree with conclusions 1 and 2 in the absence of such analyzes. The authors should make it clear that these conclusions are merely a guess based on an analysis of the literature.

Author Response

(The authors gave the same response as above.)
